# Sustained-Release Powders Based on Polymer Particles for Pulmonary Delivery of Beclomethasone Dipropionate in the Treatment of Lung Inflammation

**DOI:** 10.3390/pharmaceutics15041248

**Published:** 2023-04-14

**Authors:** Emanuela Fabiola Craparo, Salvatore Emanuele Drago, Gabriella Costabile, Maria Ferraro, Elisabetta Pace, Roberto Scaffaro, Francesca Ungaro, Gennara Cavallaro

**Affiliations:** 1Lab of Biocompatible Polymers, Department of Biological, Chemical and Pharmaceutical Sciences and Technologies (STEBICEF), University of Palermo, Via Archirafi 32, 90123 Palermo, Italy; 2National Interuniversity Consortium for Materials Science and Technology (INSTM), UdR of Palermo, Via Giusti 9, 50125 Florence, Italy; 3Laboratory of Drug Delivery, Department of Pharmacy, University of Napoli Federico II, Via Domenico Montesano 49, 80131 Napoli, Italy; 4Institute of Translational Pharmacology (IFT), National Research Council of Italy (CNR), Via Ugo La Malfa 153, 90146 Palermo, Italy; 5Department of Engineering, University of Palermo, Viale delle Scienze, 90128 Palermo, Italy; 6Advanced Technology and Network Center (ATeN Center), University of Palermo, 90133 Palermo, Italy

**Keywords:** polymeric particles, beclomethasone dipropionate (BDP), α,β-poly(N-2-hydroxyethyl)-D,L-aspartamide (PHEA), pulmonary administration, inflammation

## Abstract

Inhaled corticosteroids are the mainstay in the management of lung inflammation associated to chronic lung diseases, such as asthma and chronic obstructive pulmonary disease (COPD). Nonetheless, available inhalation products are mostly short-acting formulations that require frequent administrations and do not always produce the desired anti-inflammatory effects. In this work, the production of inhalable beclomethasone dipropionate (BDP) dry powders based on polymeric particles was attempted. As starting material, the PHEA-g-RhB-g-PLA-g-PEG copolymer was chosen, obtained by grafting 0.6, 2.4 and 3.0 mol%, respectively, of rhodamine (RhB), polylactic acid (PLA) and polyethylene glycol 5000 (PEG) on alpha,beta-poly(N-2-hydroxyethyl)DL-aspartamide (PHEA). The drug was loaded into the polymeric particles (MP) as an inclusion complex (CI) with hydroxypropyl–cyclodextrin (HP-β-Cyd) (at a stoichiometric ratio of 1:1) or as free form. The spray-drying (SD) process to produce MPs was optimized by keeping the polymer concentration (0.6 wt/vol%) constant in the liquid feed and by varying other parameters such as the drug concentration. The theoretical aerodynamic diameter (d_aer_) values among the MPs are comparable and potentially suitable for inhalation, as confirmed also through evaluation of the experimental mass median aerodynamic diameter (MMAD_exp_). BDP shows a controlled release profile from MPs that is significantly higher (more than tripled) than from Clenil^®^. In vitro tests on bronchial epithelial cells (16HBE) and adenocarcinomic human alveolar basal epithelial cells (A549) showed that all the MP samples (empty or drug-loaded) were highly biocompatible. None of the systems used induced apoptosis or necrosis. Moreover, the BDP loaded into the particles (BDP-Micro and CI-Micro) was more efficient than free BDP to counteract the effects of cigarette smoke and LPS on release of IL-6 and IL-8.

## 1. Introduction

Currently, the treatment of the chronic inflammation of diseased lungs represents a serious problem that requires personalization of pharmacological therapies and modification of the same [1]. This is associated with an increase in the incidence of side effects as well as the costs associated with either the therapy itself or the side effect management [2]. Therefore, much work is happening in the search for innovative drugs as well as on the development of new pharmaceutical dosage forms for well-established drugs in order to improve their performance in vivo [2,3,4,5]. This objective can be achieved through innovative formulations that not only allow an easier and more accepted drug administration for pulmonary diseases but also improve bioavailability at the same time or reduce dose and/or number of administrations [6,7].

Among pulmonary pathologies, chronic obstructive pulmonary disease (COPD) is currently having an increasing impact on mortality and morbidity on patients [8]. The COPD pathogenesis is multifactorial and cigarette smoke represents the major risk factor for disease development and progression [9]. Cigarette smoke causes an increase of oxidative stress, which determines airway inflammation by stimulating the release of inflammatory mediators [10]. COPD exacerbations associated with chronic bacterial infection cause an additional release of pro-inflammatory mediators such as IL-6 and IL-8 [11]. Corticosteroids are the main treatment for asthma and COPD, but severe forms are mostly resistant to their anti-inflammatory effects [12]. Oxidative stress may be a mechanism related to the corticosteroid resistance in COPD, as it increases pro-inflammatory transcription and reduces glucocorticoid-receptor-associated repressor functions [13]. In the case of poorly water-soluble corticosteroids, drug availability may be further limited by their slow dissolution in the lungs, causing clearance faster than absorption. Therefore, the optimization of corticosteroid therapy with the use of advanced pulmonary formulations, able to prolong the residence time of inhaled corticosteroids at the site of action, are highly desirable. Among them, drug encapsulation in a particulate carrier could allow the administration of lower/less frequent doses to obtain efficacy comparable with conventional therapies, with reduced systemic side effects and with a lower tendency to resistance [14,15,16,17].

In the literature, numerous attempts to improve the bioavailability of beclomethasone dipropionate (BDP) to the lungs have been already reported, i.e., by modifying the chemical–physical properties of the BDP-based powder, such as crystallinity percentage and/or drug dissolution rate [18]. In this context, satisfying results have been obtained by producing BDP nanoparticles via supercritical fluids or nanoprecipitation [19,20,21,22]. Moreover, additional advantages, such as controlled release and drug protection, have been obtained by loading the drug into nanostructured polymeric and/or lipid systems [23,24,25,26,27]. More recently, numerous BDP-based composites have been made as dry powders to be inhaled through innovative devices, to simultaneously improve the efficacy of the drug and the aerosolization properties of the formulations. Therefore, different excipients, such as sugars, polyalcohols, polymers and amino acids, have been used to produce inhalable particles via techniques such as supercritical fluids and spray drying (SD) [18,28,29,30]. In this context, some authors reported the production of inhalable matrices via supercritical assisted atomization with the aim of increasing the dissolution of BDP in biological fluids, i.e., by using cyclodextrins [31]. The obtained particles seem to be very promising as immediate-release BDP formulations for pulmonary delivery, as it was possible to obtain amorphization of the drug and very rapid dissolution kinetics. Other authors have also shown that the best inhalable BDP particles in terms of aerosol performance are obtained by SD [22]. Recently, BDP-based powder formulations were prepared by SD using different types of lactose carriers and two different dispersion media, which demonstrate that the physicochemical properties of the obtained formulations may be easily varied by altering the dispersion media composition [27]. However, in the cited notes no dissolution/release kinetics are reported in comparisons with raw drug and/or with formulations on the market.

In this work, we have chosen BDP as a corticosteroid model drug and have formulated it within polymeric particles (MPs) to be administered locally to the lungs as a dried powder. An amphiphilic fluorescent derivative of α,β-poly(N-2-hydroxyethyl)-D,L-aspartamide (PHEA) was selected as the starting polymeric material to produce the particles. Specifically, the structural and functional properties of PHEA have been modulated by functionalization with suitable amounts of polylactide (PLA) and polyethylenglycol (PEG). The potential applications of amphiphilic derivatives based on PHEA/polyesters/PEG in the biomedical field range from the production of biocompatible and biodegradable nano- and microparticles for drug delivery and theranostics up to the application in tissue engineering that has already been widely demonstrated in the last twenty years [32,33,34,35].

Dry powder production by SD of polymeric dispersions has been properly optimized in order to obtain particles with aerodynamic properties suitable to be administered as powder by using dry powder inhaler (DPI) devices. In addition, the effect of hydroxypropyl-β-cyclodextrin (HP-β-Cyd) incorporation on powder aerosolization properties, drug release kinetics and its ability to counteract cellular oxidative stress in vitro were evaluated.

## 2. Materials and Methods

### 2.1. Materials

Anhydrous N,N′-dimethylformamide (a-DMF), anhydrous dimethylacetamide (a-DMA), poly(ethylene oxide) standards, rhodamine B (RhB), D,L-poly(lactic acid) (PLA acid terminated, 10–18 kDa, ${\bar{M}}_{w}/{\bar{M}}_{n}$ = 2.22), N,N′-disuccinimidyl carbonate (DSC), beclomethasone dipropionate (BDP), 2-hydroxypropyl)-β-cyclodextrin (HP-β-Cyd, average Mw ~1380), diethylenetriaminepentaacetic acid (DTPA), RPMI amino acid solution, type II mucin from porcine stomach, egg yolk emulsion, triethylamine (TEA), diethylamine (DEA), ethyl ether, O-(2-aminoethyl)-O′-methyl poly(ethylene glycol) 5000 (H_2_N-PEG_5000_), were purchased from Sigma-Aldrich (Milan, Italy). Sodium chloride and potassium chloride were purchased from Merck (Italy). All used reagents were of analytical grade.

α,β-Poly(N-2-hydroxyethyl)-D,L-aspartamide (PHEA), PHEA-g-RhB and PHEA-g-RhB-g-PLA were obtained by synthetic procedures already reported in the literature [33,36].

^1^H-NMR spectra were registered using a Bruker Avance II-300 spectrometer, working at 300 MHz (Bruker, Milan, Italy).

PHEA ^1^H-NMR (300 MHz, D_2_O, 25 °C, TMS): δ 2.71 (m, 2H_PHEA_, -COCHC**H_2_**CONH-), δ 3.24 (m, 2H_PHEA_, -NHC**H_2_**CH_2_O-), δ 3.55 (m, 2H_PHEA_, -NHCH_2_C**H_2_**OH), δ 4.59 [m, 1H_PHEA_, -NHC**H**(CO)CH_2_-].

The weight average molecular weight (${\bar{M}}_{w})$ of PHEA was determined by SEC analysis and was found to be equal to 53.6 kDa (${\bar{M}}_{w}/{\bar{M}}_{n}$ = 1.2) [37].

PHEA-g-RhB ^1^H-NMR (300 MHz, D_2_O, 25 °C, TMS): δ 1.15 (12H_RhB_ C**H_3_**CH_2_-); δ 2.71 (2H_PHEA_ -COCHC**H_2_**CONH-); δ 3.29 (2H_PHEA_ -NHC**H**_2_CH_2_O-); δ 3.58 (2H_PHEA_ -NHCH_2_C**H_2_**O-); δ 4.65 (1H_PHEA_ -NHC**H**(CO)CH_2_-); δ 8.00–8.50 (10H_RhB_ **H**-Ar). The degree of derivatization in RhB (DD_RhB_), calculated from the ^1^H-NMR spectrum, was equal to 0.6 ± 0.05 mol%. The ${\bar{M}}_{w}\;$ of the PHEA-g-RhB used in this study was 52.5 Da (${\bar{M}}_{w}/{\bar{M}}_{n}$ = 1.6) [14].

PHEA-g-RhB-g-PLA ^1^H-NMR (300 MHz, [D7]DMF, 25 °C, TMS): δ 1.15 (m, 12H_RhB_ C**H**_3_CH_2_-); δ 1.3 and δ 1.7 (2d, 3H_PLA_ –[OCOCH(C**H_3_**)]_194_-); δ 2.8 (m, 2H_PHEA_ -COCHC**H_2_**CONH-); δ 3.3 (t, 2H_PHEA_ -NHC**H_2_**CH_2_O-); δ 3.59 (t, 2H_PHEA_ -NHCH_2_C**H_2_**O-); δ 4.2-4.5 and δ 5.1-5.5 (m, 1H_PLA_ –[OCOC**H**(CH_3_)]_194_-) and δ 4.8 (m, 1H_PHEA_ -NHC**H**(CO)CH_2_-); δ 7.0-8.0 (m, 10H_RhB_ **H**-Ar). The degree of derivatization in PLA (DD_PLA_), calculated from the ^1^H-NMR spectrum, was equal to 2.4 ± 0.06 mol% [34]. ${\bar{M}}_{w}\;$ of the PHEA-g-RhB-g-PLA graft copolymer was found to be 180.0 kDa (${\bar{M}}_{w}/{\bar{M}}_{n}$ = 1.45).

### 2.2. Synthesis and Characterization of PHEA-g-RhB-g-PLA-g-PEG_5000_

H_2_N-PEG_5000_-OCH_3_ was properly grafted on PHEA-g-RhB-g-PLA by following an adapted procedure [33]. In detail, to an organic dispersion of PHEA-g-RhB-g-PLA (64 mg/mL in a-DMA) at 40 °C, DEA and DSC were added, according to R_1_ = 0.1 and R_2_ = 1, R_1_ being the molar ratio between the DSC and the repeating units (RUs) of PHEA-g-RhB-g-PLA carrying hydroxyl groups and R2 the molar ratio between DEA and DSC. The reaction mixture was kept at 40 °C for 4 h and then added to a solution of H_2_N-PEG_5000_-OCH_3_ (24 mg/mL in a-DMA) in a such way as to have R_3_ = 0.075, R_3_ being the molar ratio between the amine terminal groups on H_2_N-PEG_5000_-OCH_3_ and RUs PHEA-g-RhB-g-PLA. The obtained mixture reaction was left at 25 °C for 18 h under argon and continuous stirring, then dialysed against distilled water (Spectra/Por^®^ Standard RC tubing; MWCO 12-14 kDa) and lyophilised. The product was obtained with a yield of 340 wt% based on the starting PHEA-RhB weight.

PHEA-g-RhB-g-PLA-PEG_5000_ ^1^H-NMR (300 MHz, [D7]DMF, 25 °C, TMS): δ 1.15 (m, 12H_RhB_ C**H_3_**CH_2_-); δ 1.3 and δ 1.7 (2d, 3H_PLA_ –[OCOCH(C**H_3_**)]_194_-); δ 2.8 (m, 2H_PHEA_ -COCHC**H_2_**CONH-); δ 3.3 (t, 2H_PHEA_ -NHC**H_2_**CH_2_O-); δ 3.59 (t, 2H_PHEA_ -NHCH_2_C**H_2_**O-); δ 3.7 (t, 4H_PEG_ -[C**H_2_**C**H_2_**O]_114_-); δ 4.2–4.5 and δ 5.1–5.5 (m, 1H_PLA_ –[OCOC**H**(CH_3_)]_194_-) and δ 4.8 (m, 1H_PHEA_ -NHC**H**(CO)CH_2_-); δ 7.0–8.0 (m, 10H_RhB_ **H**-Ar). ${\bar{M}}_{w}$ of the PHEA-RhB-PLA graft copolymer was found to be 198.0 kDa (${\bar{M}}_{w}/{\bar{M}}_{n}$ = 1.34) [33]. The degree of derivatization in PEG (DD_PEG_), determined from the ^1^H-NMR spectrum, was equal to 3.0 ± 0.05 mol%.

### 2.3. Preparation of BDP/HP-β-Cyd Inclusion Complex (CI)

A complex of BDP with HP-β-Cyd (CI) was prepared by adding to an aqueous HP-β-Cyd dispersion (10% *w*/*v*) an excess amount of BDP powder (0.4% wt/v). The obtained suspension was left at 37 °C for 24 h, alternating 5 sonication cycles at 35kHz for 5 min (Bandelin electronic sonicator, Berlin, Germany). Then, the dispersion was filtered through a 0.22 μm membrane filter and the concentration of BDP in the filtrate was analyzed using high-pressure liquid chromatography (HPLC). Moreover, the filtrate was freeze-dried and the DL% (expressed as the weight percent ratio between the loaded BDP and the dried CI) was quantified by dispersing the sample in MeOH. After centrifugation and filtration, the supernatant was analyzed by HPLC. In parallel, the intrinsic solubility of BDP in water was determined by incubating BDP powder alone in the same experimental conditions described above.

### 2.4. BDP Quantification

BDP quantification was carried out by HPLC using a Waters Breexe System Liquid Chromatograph equipped with a Waters 717 Plus Autosampler (40 μL injection volume) and a Shimadzu UV−vis HPLC detector online with a computerized workstation [18]. The column was a Luna^®^ C18 (Phenomenex, Torrance, BO, Italy), the mobile phase was a mixture of methanol:water 70:30 *v*/*v* with a flow rate of 1 mL/min, the column temperature was 25 °C and the detection wavelength was 238 nm. The obtained peak areas at 8 min were compared with a calibration curve obtained by plotting areas versus standard solution concentrations of BDP in methanol in the range of 0.01–0.1 mg/mL (y = 69.786x, R_2_ = 0.998).

To evaluate the drug loading (DL%), 5 mg of each MP sample was weighed, dissolved into 200 μL of DMSO and then 4.8 mL MeOH was added. After centrifugation and filtration, the supernatant was analyzed by HPLC. Results were expressed as DL%, that is the weight percent ratio between the loaded BDP and the dried system (CI or particle sample). Entrapment efficacy (EE%) was expressed as the weight percentage ratio between the amount of BDP actually entrapped into the sample and the theoretical one.

### 2.5. General Procedure for Preparation of Dry Powders

The spray-drying process was performed using a Büchi Nano Spray Dryer B-90 (Büchi, Flawil, Switzerland) with a Büchi B-90 dehumidifier (Büchi, Switzerland). Instrument experimental process parameters were fixed at an inlet temperature of 55 °C (outlet temperature was controlled in the range of 30–32 °C), pump flow at 66%, spray at 78%, internal pressure at 25 hPa and a gas flow rate of 110–120 L/min using compressed air. The medium nebulizer was used. Other parameters were: the acetone:ethanol v:v ratio in the liquid feed was equal to 7.9:2.1 or 7.5:2.5, PHEA-g-RhB-g-PLA-g-PEG_5000_ graft copolymer concentration at 0.6 *w*/*v*%, HP-β-Cyd concentration at 0.03 or 0.06 *w*/*v*%, CI concentration at 0.04 *w*/*v*% and BDP concentration at 0.01 or 0.1 *w*/*v*%. After SD, each powdered sample was gently recovered from the electrostatic collector by scratching and stored in the fridge before use. The qualitative–quantitative composition of the liquid feed for the SD process for each MP sample is reported in Table 1.

### 2.6. Dry Powder Characterization

#### 2.6.1. Morphological Characterization and Geometric Diameter (d_g_)

Particle morphology was investigated by scanning electron microscopy (SEM) by putting a sample on double-sided adhesive tape, previously applied on a stainless-steel stub. The latter was then sputter-coated with gold prior to microscopy examination and observed by using a Phenom™ ProX Desktop SEM microscope. The ImageJ program was used to calculate the average geometric diameter (d_g_) of each sample from SEM images by analyzing a sufficiently representative number to constitute a certain datum (>500 particles).

#### 2.6.2. ζ Potential Measurements

The ζ potential values (mV) of each sample were determined in bidistilled water from electrophoretic mobility using the Smoluchowski relationship by using a Zetasizer NanoZS (Malvern Instruments, Worcestershire, UK) instrument. Analyses were performed in triplicate.

#### 2.6.3. Tapped Density

Tapped density of MP samples was measured by the syringe method [38,39]. Each powder was filled in a 1 mL graduated syringe and the powder weight required to fill the syringe was recorded to calculate the bulk density. Then, the tapped density (ρ_tapp_) was calculated from the volume value determined by tapping the syringe onto a level surface at a height of 2.5 cm 100 times and repeating the tapping until a constant volume was reached. Each measurement was performed in triplicate. 

#### 2.6.4. Theoretical Aerodynamic Particle Diameter

The theoretical aerodynamic diameter (d_aer_) was calculated from the ρ_tapp_ and the d_g_ by using the following equation [40,41].
(1)$$d_{\mathrm{aer}}=d_{g}\;\sqrt{\frac{\rho_{\mathrm{tapp}}}{\chi \;\times {\;\rho }_{0}}}$$
where:

d_aer_ = aerodynamic diameter, μm;

d_g_ = geometric diameter, μm;

ρ_tapp_ = tapped density, g/cm^3^;

χ = form factor (=1 for spherical particles);

ρ_0_ = unit density.

#### 2.6.5. In Vitro Aerosol Performance

The aerosolization properties of the dry powders were tested in vitro after delivery from breath-activated reusable DPIs working with a single unit capsule containing the dry powder and using a Next Generation Impactor (NGI) (Copley Scientific, Nottingham, UK) according to *Ph. Eur. 11th Ed*. Three devices with different resistances to the airflow were tested: the low-resistance Breezehaler (Plastiape, Osnago, Italy), the medium-resistance DPI Turbospin^®^ (PH&T Pharma, Milan, Italy) and the high-resistance DPI HandiHaler^®^ (Boehringer Ingheleim, Ingelheim am Rhein, Germany) [42]. For each test, a hard gelatin capsule (size 2, Capsugel, Greenwood, SC, USA) was filled with about 15 mg of the powder and placed in the DPI. The capsule was then pierced and the liberated powder drawn through the NGI operated at 39, 60 or 90 L/min.

The powder deposited on the seven NGI collection cups, in the induction port and in the micro-orifice collector (MOC), was quantitatively recovered. In a first set of experiments, the amount of powder was collected by dissolution in an appropriate amount of DMF and analyzed by spectrofluorimetric analysis at λ_ex_ = 560 nm and λ_em_ = 590 nm (GloMax^®^ Explorer Multimode Microplate Reader, Promega, Milan, Italy) for the PHEA-g-RhB-g-PLA-g-PEG5000 graft copolymer content. A calibration curve was derived from serial dilutions of a standard solution (1 mg/mL) of fluorescent MPs in DMF (0.08–1.0 mg/mL concentration range, R^2^ ≥ 0.99). In a second set of experiments, the amount of BDP deposited was calculated. Briefly, samples were collected in an appropriate amount of water, then frozen and freeze-dried to allow the extraction of BDP through DMSO and its quantification by HPLC as reported above. The emitted dose (ED) was measured as the difference between the total amount of powder initially placed and the amount remaining in the capsule. Upon emission, the experimental mass median aerodynamic diameter (MMAD_exp_) and the geometric standard deviation (GSD) were calculated according to the *Ph. Eur*., deriving a plot of the cumulative mass of powder retained in each collection cup (expressed as a percentage of total mass recovered in the impactor) versus cut-off diameter of the respective stage. The MMAD_exp_ was determined from the graph as the particle size at which the line crosses the 50% mark; the GSD was defined as
GSD = (Size X/Size Y)^1/2^
where size X was the particle size at which the line crosses the 84% mark and size Y the size at which it crosses the 16% mark.

The fine particle fraction (FPF) was calculated as the percentage ratio between the amount of dry powder deposited on stage 3 through MOC and the total amount loaded into the capsule. The respirable fraction (RF) was calculated as the percentage ratio between the amount of dry powder deposited on stage 3 through MOC and the amount of dry powder deposited throughout the NGI (emitted dose).

### 2.7. Drug Release in Artificial Mucus (AM)

AM was prepared by dispersing 50 mg of mucin, 0.060 mg of DTPA, 0.2 mL of RPMI 1640 Amino Acid Solution, 50 μL of egg yolk emulsion, 50 mg of NaCl and 22 mg of KCl in 10 mL of bidistilled water. This dispersion was allowed to equilibrate at 25 °C for 2 h, and experiments were performed within 48 h of AM preparation.

The BDP release assay from BDP-Micro and CI-Micro samples was performed by using vertical Franz-type diffusion cells in the presence of AM. In detail, experiments were conducted by placing in the donor chamber a BDP-Micro and CI-Micro dispersion in AM (500 μL) at the concentration of BDP equal to 0.4 mg/mL. Control experiments were conducted by placing in the donor compartment a Clenil^®^ dispersion in AM (500 μL) (total BDP concentration equal to 0.4 mg/mL). The acceptor chamber was filled with 5 mL of DPBS and a cellulose acetate membrane (size pores: 0.45 μm), previously set in DPBS overnight, was applied between the two compartments. The system was kept at a constant 37 °C by recirculation of water from a thermostatically controlled bath under continuous stirring at 180 rpm. At defined time intervals (0, 10, 20, 30, 40, 60, 120, 180, 240, 300 and 360 min), aliquots (400 μL) were removed from the acceptor chamber and replaced with the same volume of DPBS. The withdrawn samples were freeze-dried, treated with the same volume of methanol, filtered and injected using the HPLC method described above. All the experiments were run in triplicate.

### 2.8. Biological Characterization

#### 2.8.1. Cell Lines

Biological evaluations were performed on 16HBE, an immortalized normal bronchial epithelial cell line [43], and A549, an adenocarcinomic human alveolar basal epithelial cell line. 16HBE and A549 cells were maintained in a humidified atmosphere of 5% CO_2_ in air at 37 °C, both were cultured as adherent monolayers. Eagle’s minimum essential medium (MEM), supplemented with 10% heat-inactivated (56 °C, 30 min) foetal bovine serum (FBS), 1% MEM (non-essential amino acids), 2 mM L-glutamine and 0.5% gentamicin (all from Euroclone), was used for culturing 16HBE cells. RPMI-1640 medium supplemented with heat-deactivated (56 °C, 30 min) 10% FBS, 1% streptomycin and penicillin, 1% MEM (non-essential aminoacids) and 2mM L-glutamine (all from Euroclone) was used for culturing A549 cells [44].

#### 2.8.2. Preparation of Cigarette Smoke Extracts (CSE)

Kentucky 3R4F research-reference cigarettes (The Tobacco Research Institute, University of Kentucky) without filters were used. Cigarette smoke extract (CSE) was prepared using a peristaltic pump, Watson-Marlow 323 E/D (Rotterdam, The Netherlands). Briefly, each cigarette was smoked for 5 min and one cigarette was used to generate 10 mL of CSE-PBS solution in PBS. The obtained CSE solution was filtered through a 0.22 μm pore filter to eliminate bacteria and large particles. The CSE solution was used within 30 min of preparation. This solution was set as 100% CSE and was opportunely diluted to obtain the desired concentration for each experiment. The concentration of CSE was verified spectrophotometrically, measuring the OD at a wavelength of 320 nm as previously described [45]. The pattern of absorbance among the different batches showed small differences as the mean OD of the different batches was 1.37 ± 0.16. The presence of contaminating LPS in undiluted CSE solution was assessed by a commercially available kit (Cambrex Corporation, East Rutherfort, NJ, USA) and was below the detection limit of 0.1 EU/mL.

#### 2.8.3. Stimulation of Cell Lines

Cells were grown until 80–90% confluency in 96-well plates to evaluate cell viability, in 12-well plates to evaluate cell apoptosis or in 6-well plates to evaluate the release of cytokines. After seeding, cells were treated in medium 1% FBS for 24 h with BDP in dispersion or incorporated into the MP systems (BDP-Micro or CI-Micro) (10^−6^, 10^−8^ and 10^−10^ M), with or without CSE 10% and lipopolysaccharide (LPS) 1 μg/mL (Sigma-Aldrich, St Louis, MO, USA). The sample empty Micro was tested as well and used as a reference. The free drug or samples were added 1 h before CSE and LPS cell stimulation. At the end of stimulation, cells were collected for further evaluation. The concentrations of CSE and LPS and the time of incubation were selected on the basis of previous findings [44]. Each experiment was performed in triplicate.

#### 2.8.4. Cell Viability Assay

The biocompatibility of MP samples was evaluated by means of the CellTiter 96^®^ Aqueous One Solution Cell Proliferation Assay (Promega, Madison, WI, USA), a colorimetric method for determining the number of viable cells. The kit reagent contains MTS [3-(4,5-dimethylthiazol-2-yl)-5-(3-carboxymethox-yphenyl)-2-(4-sulfopheyl)2H-tetrazolium], which is reduced by cells into crystals that are soluble in the culture medium. The quantity of formazan produced by the cells, measured at 490 nm, is directly proportional to the number of metabolically active cells.

At the end of the treatment with BDP (free or loaded into MP (BDP-Micro or CI-Micro) samples), 20 μL of CellTiter 96^®^ AQueous One Solution reagent was added to each well and the plates were incubated for 20 min at 37 °C and 5% CO_2_. The absorbance was measured using a microplate reader at 490 nm (Microplate reader SPECTROstar-Nano (BMG-Labtech, Allmendgrün, Ortenberg). The results are expressed as percentage of viability compared with untreated cells (100% viability).

#### 2.8.5. Cell Apoptosis by the Annexin V Binding Method

Cell apoptosis in the presence of BDP in dispersion or polymeric particle systems was evaluated by staining with annexin V–fluorescein isothiocyanate and propidium iodide (PI) using a commercial kit (Bender MedSystem, Vienna, Austria) following the manufacturer’s directions. Flow cytometry analyses were performed on CytoFLEX (BeckmanCoulter, Brea, CA, USA).

#### 2.8.6. IL-8 and IL-6 Release

The release of IL-8 and IL-6 was evaluated by the ELISA method (enzyme-linked immunosorbent assay) (R&D Systems, Minneapolis, MN, USA) according to the manufacturer’s instructions.

#### 2.8.7. Statistical Analysis

Data were expressed as mean ± SD and analyzed by analysis of variance (ANOVA) followed by Bonferroni’s correction. A *p* value less than 0.05 was considered to be statistically significant.

## 3. Results

The incidence of chronic inflammatory pulmonary diseases, such as asthma and chronic obstructive pulmonary disease (COPD), is currently growing, as is the search for more effective and selective drugs as well as the search for more effective and accepted therapies with conventional drugs [8]. This latter strategy involves optimizing the bioavailability of drugs already known by means of new formulations as well as new devices for their administration or both.

With this goal, in this work we have dealt with the realization of a particle-based formulation for beclometasone dipropionate (BDP), to be locally administered to the lung as dry powder, to allow the optimization of the drug bioavailability and a reduction of the systemic side effects. In detail, the formulation was made up of polymeric particles obtained by spray drying (SD). The SD process was chosen because it represents a valid process for easily obtaining systems suitable for pulmonary administration [46,47]. SD, combined with microfluidic anti-solvent precipitation and high-pressure homogenization (HPH), has already been described as a valid method to prepare ultrafine BDP particles for dry powder inhalation (DPI) administration, with a higher aerosol performance than raw BDP [22]. However, no modification of the dissolution profile was still demonstrated.

Here, the production of BDP-loaded particles should have a double impact: on the one hand, an inhalation powder to be administered locally is obtained, on the other hand, the modification of the drug release profile having an impact on the drug effects. Moreover, the drug was loaded into the MPs in the form of an inclusion complex (CI) with hydroxypropyl-β-cyclodextrin (HP-β-Cyd) or in the free form, in both cases with the purpose to modulate the release profile in biological fluids, as well as the local effect of the drug via increased solubility [48]. The improved performance of spray-dried excipients in the presence of HP-β-Cyd, thanks to its effects on the aerosolization properties of the resulting powders, is already widely demonstrated [49].

To obtain the particles, the first step of the work was the synthesis and characterization of a polymeric material that constituted the particle matrix. A biocompatible and highly versatile material was chosen, such as a derivative of the α,β-poly(N-2-hydroxyethyl)-DL-aspartamide (PHEA), which was synthesized by reactions with rhodamine (RhB), polylactic acid (PLA) and with polyethylene glycol 5000 (PEG) [33,34,36]. As reported for other PHEA-based copolymers, each functionalization on the PHEA backbone was carried out to make the copolymer highly performing: the RhB makes it fluorescent, PLA makes it insoluble in aqueous fluids, and therefore allows it to be used to produce systems for the controlled release of drugs, and the PEG allows the hydrophilicity/lipophilicity features, as well as improves its biocompatibility [33]. The pegylation of PHEA-g-RhB-g-PLA and the chemical structure of PHEA-g-RhB-g-PLA-g-PEG are depicted in Figure 1.

The resulting copolymer was characterized by ^1^H-NMR analysis to determine the degree of derivatization (DD) with RhB, PLA and PEG, which were found to be equal to 0.6, 2.4 and 3.0 mol%, respectively. Moreover, a SEC analysis was performed to determine the average molecular weight (${\bar{M}}_{w}$) and the PDI (${\bar{M}}_{w}/{\bar{M}}_{n}$) of PHEA-g-RhB-g-PLA-g-PEG graft copolymer, which were found to be 198 kDa and 1.34, respectively. This analysis demonstrated the absence of degradation of the resulting copolymer due to the various synthetic steps, the value obtained being very similar to the weight of the theoretical value.

Among the organic solvents able to dissolve the copolymer, acetone was chosen as the main liquid feed, being widely reported in the literature to be one of the best performing solvents for the SD process to obtain polymeric particles as a dry powder from various starting materials [47].

Moreover, the inclusion complex (CI) between BDP and HP-β-Cyd was prepared at a stoichiometric ratio of BDP:HP-β-Cyd equal to 1:1 (determined by HPLC analysis, as described in the experimental part), corresponding to a DL% (i.e., the weight percent ratio between the drug amount found and the CI) equal to 25 wt%.

At this point, the MP production process has been optimized in order to obtain particles of adequate size and containing BDP engaged in the inclusion complex or free. For this purpose, some parameters have been made to vary. The composition of the liquid feed to produce each MP sample, in terms of polymer, drug and/or HP-β-Cyd amount, and the volume ratio between acetone and water is reported in Table 1 (see the experimental part). The SD process has been schematized in Figure 1.

Each particulate sample, immediately after being produced and recovered from the electrostatic collector, was characterized by dimensional analysis using the ImageJ program on images obtained by SEM analysis. The images of the empty samples (Empty Micro), those containing the CI (CI-Micro) and those containing the drug (BDP-Micro_high_ and BDP-Micro), are shown in Figure 2, whereas the mean size and standard deviations for all the obtained samples are reported in Table 2. As can be seen, the particles have a spherical morphology and a fairly low polydispersity. Moreover, all particle samples are negatively charged, with no significant differences among them, as evidenced by the ζ potential values ranging between −9.50 and −16.00 mV, also reported in Table 2.

Particle size is a crucial property of any inhaled aerosol and influences its performance and deposition in the respiratory airways. Samples were initially produced in the absence of drug to optimize the blank matrix. The sample Empty Micro, obtained using a polymer dispersion in an acetone:water mixture 7.9:2.1 vol/vol, showed a mean size equal to 1.36 ± 0.19 μm. Cyd-Micro and Cyd-Micro_high_ samples were produced by entrapping the HP-β-Cyd into the polymeric matrices to evaluate the effect of the HP-β-Cyd concentration on the MP mean diameter. Cyd-Micro was obtained by using as a mixture of acetone:water and a concentration of HP-β-Cyd equal to 7.9:2.1 vol:vol and 0.03 wt/vol%, respectively, whereas the sample Cyd-Micro_high_ was a mixture 7.5:2.5 vol:vol and 0.06 wt/vol%, respectively. By dimensional analysis, it was found that the increase in the water content in the liquid feed, justified by the need to increase the quantity of Cyd in the obtained particles, led to MPs with a significantly lower average diameter being obtained. Specifically, the mean size was equal to 1.43 ± 0.24 μm and 1.07 ± 0.24 μm for Cyd-Micro and Cyd-Micro_high_ samples, respectively (*p* < 0.05). This mean size was at the limit of the accepted range for pulmonary administration; we chose to carry out the subsequent experiments by using the solvent mixture at the lowest water content level and a Cyd concentration equal to 0.03 wt/vol%. The subsequent experiment was carried out by dispersing the inclusion complex (CI) in the polymeric dispersion and, after SD, the obtained MP sample (CI-Micro) showed a mean diameter of 1.35 ± 0.25 μm. The particulate samples, named BDP-Micro_high_ and BDP-Micro, were obtained by dissolving in the chosen polymer acetone/water dispersion concentrations of BDP equal to 0.1 and 0.01 wt/vol%, which showed mean size equal to 1.26 and 1.30 μm, respectively.

The particle samples containing BDP, as CI or in the free form, were characterized by HPLC analysis to determine the drug content, expressed as drug loading (DL%) (i.e., the weight percentage ratio between the drug amount found and the MPs), and entrapment efficiency (EE%) (i.e., the weight percentage ratio between the amount of drug loaded into MPs and the theoretical value). To calculate the EE% of the CI-Micro sample, the theoretical drug amount present in the CI (stoichiometric ratio between BDP and HP-β-Cyd equal to 1:1) added to the liquid feed was considered. The obtained data are reported in Table 3.

As expected, as the BDP amount in the liquid feed increases (BDP-Micro_high_), the DL% of the obtained MP samples increases, whereas the samples obtained from comparable quantities of drug (BDP-Micro and CI-Micro) show comparable DL values.

The success of any inhalation therapy based on DPIs primarily depends on the flow properties and consequent aerodynamic behaviour of the dry powders. Thus, the effect of MP composition on particle density and aerodynamic diameter was investigated. The values of tapped density (ρ_tapp_) were calculated using the syringe method (Appendix A) [39]. Data show that there are significant differences in the ρ_tapp_ values depending on the composition of the liquid feed. In particular, in the presence of free BDP instead of CI, as well as with an increased amount of BDP, the ρ_tapp_ increases. The theoretical aerodynamic diameter (d_aer_) for each MP sample was calculated from the mean geometric diameter (d_g_) according to Equation (1). As can be seen, the calculated d_aer_ values among the samples are comparable and do not differ significantly from each other. In all cases, these are all potentially suitable values for pulmonary administration by inhalation of dry powder.

The ability of the developed dry powders to deposit in the deep airways was also investigated using the Next Generation Impactor (NGI) as described in the *Ph. Eur. 11th Ed*. In a first step, a compelling study of the aerosolization properties of the dry powders was performed on fluorescent MPs by quantifying the fluorescence of the copolymer with which the particles were produced. The powder was emitted from DPIs characterized by progressively higher intrinsic resistance to the airflow (i.e., Breezhaler^®^, Turpospin^®^ and HandiHaler^®^); therefore, the NGI was operated at different flow rates, useful to draw 4.0 L of air at a pressure drop of 4 kPa through the inhaler according to *Pharmacopoeia*. The results are shown in Appendix A as cumulative mass of powder recovered as a function of the cut-off diameter of the NGI (Appendix A) and percentage of powder recovered from the throat, on the seven NGI cups and in MOC (i.e., NGI deposition pattern) (Appendix A). The NGI deposition pattern of the dry powders varied as a function of the DPI resistance. When the high-resistance HandiHaler^®^ was tested, less than 5% of the emitted dose was recovered from cup 3 to MOC (i.e., respirable fraction or RF). Little improvement in the aerosol performance was achieved with Turpospin^®^, a medium-resistance DPI, showing an RF of around 14% (Appendix A). Nonetheless, the developed dry powders showed the best aerosol performance when using Breezhaler^®^, a low-resistance DPI operated at a flow rate of 90 L/min, allowing an increase in the RF up to an estimated value of 20%. This is supposed to be even higher than 20%, considering the fraction of dry powder deposited on cup 2 (6.48–3.61 μm) with an aerodynamic diameter lower than 5 μm. The results are not surprising, since powders with small geometric diameter usually suffer from poor flowability and dispersibility due to the increased ratio of interparticle forces to dispersive forces [18]. This issue can be partly overcome by increasing the air flow through the inhaler, which increases the shear force and particle-device collision [50], as in the case of the Breezhaler (90 L/min). It should be underlined that the low resistance of this DPI results in a smaller inspiratory effort being required by patients for its effective use. This allows the device to be effectively used across a wide age range of patients and disease severities [51].

In light of these preliminary data, further analyses were performed to evaluate the amount of BDP deposited throughout the NGI upon delivery through Breezhaler^®^. The results are shown in Figure 3 as cumulative mass of BDP-Micro or CI-Micro recovered as a function of the cut-off diameter of the NGI (Figure 3A) and percentage of powder recovered from the throat, on the seven NGI cups and in MOC (i.e., NGI deposition pattern) (Figure 3B). As can be seen, BDP-Micro and CI-Micro displayed very different behaviours upon aerosolization.

Though the percentage of CI-Micro deposited in the throat was higher than that of BDP-Micro (32.5 ± 2.2% and 12.9 ± 5.7%, respectively), an amount of BDP-Micro as high as 65% was recovered from Cup 1. A 3-fold reduction in this value was observed for CI-Micro, displaying the highest RF (36.2 ± 1.9% versus 9.6 ± 1.2% of BDP-Micro). Upon addition of HP-β-Cyd, the value of MMAD_exp_ decreased as well, with CI-Micro and BDP-Micro displaying a MMAD_exp_ of 4.15 ± 0.108 μm and 12.9 ± 0.490 μm, respectively. Thus, by the same geometric diameter (Table 2) and DPI resistance, the incorporation of HP-β-Cyd inside the MPs clearly contributed to powder flowability.

Several works report attempts to formulate BDP, either alone or in the form of complexes with different Cyds, in matrices made of pharmaceutical excipients by SD [22,48]. However, no data relating to the effects of complexation/excipients on the dissolution/release profile in biological fluids is reported.

In our work, the polymer matrix plays a fundamental role in modulating the drug release profile of the fluid it comes into contact with. Therefore, to evaluate the release profile of BDP from MPs, and whether this is influenced by the involvement of the drug in the inclusion complex versus the free form, the vertical diffusion Franz cell was used in the presence of artificial mucus in the donor compartment to mimic the lung conditions [52]. Specifically, the cumulative release profile of BDP from CI-Micro and BDP-Micro samples (containing a comparable amount of BDP) was evaluated. For comparison, the same determination was carried out using the commercial formulation of the BDP, namely Clenil^®^, a nebulizer suspension of BDP administered to reduce swelling and irritation in the walls of the airways, thus alleviating breathing problems, e.g., in patients with asthma.

The cumulative release profiles are reported in Figure 4.

Looking at the graph, it is evident that BDP from both MP samples shows a controlled release profile that is significantly higher than the commercial Clenil^®^ formulation. Specifically, after 6 h of incubation, the BDP release from the CI-Micro is significantly higher than that from the BDP-Micro (6.7 wt% vs. 3.7 wt%) (*p* < 0.001), and at the same time it is more than tripled that for Clenil^®^ (1.5 wt%) (*p* < 0.001). Therefore, the involvement of the drug in the inclusion complex with HP-β-Cyd apparently increases the dissolution rate of the drug in biological fluids. Therefore, both formulations containing BDP show a modified release profile of the drug in biological fluids, compared with that obtained with the commercial formulation. This fact may be mainly due to the presence of BDP in the amorphous state, both in the free form (in the case of the BDP-Micro sample) or involved in the inclusion complex (in the case of the CI-Micro sample), in the particle-based formulations. With the release profiles of BDP being different with respect to the commercial formulation due to the entrapment into the polymeric carrier, it could determine also an optimization of the efficacy of the drug. For this reason, in vitro tests were carried out on lung cell lines in order to determine the effect of the BDP incorporated in the polymeric particles compared with the free drug.

The first step required for the use of these particles as new drug delivery systems at the lung level is the verification of the biocompatibility at the cellular level. Therefore, the effect of empty or drug-loaded MP samples on cell viability was tested in vitro by using bronchial epithelial cells (16HBE) and human alveolar basal epithelial cells (A549).

These cells have been chosen as a model because they represent the first pulmonary barrier against environmental and pollutant inhaled substances as well as the first target of inhaled drugs, 16HBE and A549, respectively, and are a representative model of the epithelium of the central and distal airways. Specifically, 16HBE is a cell line that retains the differentiated morphology and function of normal airway epithelial cells and is used to study the functional properties of bronchial epithelial cells in inflammation and repair processes, whereas the A549 is an ideal model of type II alveolar cells to study pulmonary drug delivery mechanisms [53]. The metabolic activity of these cells was measured to evaluate the biocompatibility of each sample using the MTS viability assay. The cells were treated for 24 h with free BDP in dispersion or incorporated into the MP systems (CI-Micro or BDP-Micro). Empty Micro were tested as well under the same conditions to verify the effect of the vehicle on cell viability. As can be seen in Figure 5, BDP, either free or incorporated in the MP systems, was highly biocompatible; the viability was never lower than 90% for both cell types at all concentrations tested. The same behaviour was obtained with the empty formulation, confirming the high biocompatibility of the MP systems. This preliminary test confirmed the good properties of these systems and their suitability for further pharmacological tests.

The concentration 10^−8^ M was selected because it was more effective on the basis of preliminary dose-response experiments and a previous report [54]. Only this concentration was used for cytokine release tests, and cell apoptosis/necrosis were evaluated by annexin/propidium method [55]. As can be seen in Figure 6, none of the systems used induced apoptosis or necrosis, confirming their non-toxicity.

In order to evaluate the effect of BDP-loaded particle samples on the release of IL-8 and IL-6 from 16HBE and A549 cells, both cell lines were stimulated with cigarette smoke and lipopolysaccharides (from Gram-negative bacteria), simulating the conditions of a smoker subject exposed to microbial infection. IL-6 and IL-8 are pro-inflammatory cytokines that contribute to disease severity in asthma and COPD and are therefore good therapeutic targets [56]. Both CI-Micro or BDP-Micro are efficient in counteracting the effects of cigarette smoke and LPS on the release of IL-6 in A549 cells (Figure 7B) and IL-8 in 16HBE and A549 cells (Figure 8A,B). Moreover, the BDP-Micro seems to be more efficient than either free BDP or CI-Micro in counteracting the effects of cigarette smoke and LPS on the release of IL-6 and IL-8 in 16HBE and A549 cells.

## 4. Conclusions

In conclusion, polymeric particles (MP) were produced as carriers for local administration of beclomethasone dipropionate (BDP) to the lung. Specifically, these particles were obtained starting from a biocompatible graft copolymer by means of spray drying (SD). The process parameters as well as the qualitative–quantitative composition of the feed liquid have been suitably optimized in order to obtain MP samples of >1 μm and low polydispersity.

Two MP samples were selected with a comparable amount of loaded BDP in the form of an inclusion complex with hydroxypropyl-β-cyclodextrin (HP-β-Cyd) or as free form. Such incorporation into MP samples in both cases determines a modification of the drug release profile in fluids mimicking biological ones, as well as an improvement of the aerosolization performance when the most suitable aerosolization device is selected.

Using in vitro studies on bronchial epithelial cells (16HBE) and human alveolar basal epithelial cells (A549), the good biocompatibility and a significantly higher efficacy of both these formulations compared with free BDP were demonstrated, with them being able to counteract the effect of cigarette smoke and LPS on the release of pro-inflammatory cytokine such as IL-8 and IL-6.

The results in terms of aerosolization properties, drug release profile and anti-inflammatory activity of entrapped BDP, suggest the potential use of these BDP-loaded particles, and particularly CI-Micro, as new innovative drug delivery systems for the treatment of pulmonary diseases characterized by high oxidative stress and inflammation.

## Data Availability

No supplementary data available.

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
