# Peer review of "Sustained-Release Powders Based on Polymer Particles for Pulmonary Delivery of Beclomethasone Dipropionate in the Treatment of Lung Inflammation"

_pharmaceutics, 2023, doi:10.3390/pharmaceutics15041248_

Round 1

Reviewer 1 Report

This is a well-done study about sustained-release powders based on polymer particles for pulmonary delivery of beclomethasone dipropionate in the treatment of lung inflammation. I will suggest it for publication after the following minor points are fixed.

1. Line 72-74, several studies (Langmuir 35 (5), 1273-1283; Pharmaceutics 2022, 14(12), 2770) should be included to support such a claim.

2. The information about the PDI of PLA should be added.

3. The quality of figures 3, 4, and 6 should be improved to a higher level.

4. The zeta potential of the particles should be measured.

Author Response

Answers to Review 1

  1. Line 72-74, several studies (Langmuir 35 (5), 1273-1283; Pharmaceutics 2022, 14(12), 2770) should be included to support such a claim.

As the reviewer suggests, the authors have added some bibliographic references at page 2, line 29 of the revised version of the manuscript), including those indicated by the reviewer [Singh, A.P. et al. Signal Transduct. Target. Ther. 2019, 4, 1–21; Ulldemolins, A. et al. Cancer Drug Resist. 2021, 4, 44–68; Lin, W. et al. 2019, 35, 1273–1283; AbouAitah, K. et al. Pharmaceutics 2022, 14, 2770].

  1. The information about the PDI of PLA should be added.

As the reviewer suggests, the authors have inserted the PDI value [ = 2.22] at page 2, line 87, experimentally determined in the same conditions used for the other synthesized copolymers.

  1. The quality of figures 3, 4, and 6 should be improved to a higher level.

According to the reviewer’s suggestion, the authors have improved the quality of the figures 3, 4 , and 6.

  1. The zeta potential of the particles should be measured.

As the reviewer suggests, the authors have carried out measurements of zeta potential values of obtained particles. Therefore, they have added the obtained values ± standard deviation in the Table 2, and the following paragraph at page 5, lines 28-32 of the revised version of the manuscript: “2.6.2 z potential measurements. z potential values (mV) of each sample were determined in bidistilled water from electrophoretic mobility using the Smoluchowski relationship, by using a Zetasizer NanoZS (Malvern Instruments, Worcestershire, UK) instrument. Analyses were performed in triplicate.”. They have also added at page 11, lines 12-14, the following sentences: “Moreover, all particle samples are negatively charged, with not significant differences among them, as evidenced by the z potential values ranging between -9.50 and -16.00 mV, also reported in Table 2.  

Reviewer 2 Report

The manuscript entitled “Sustained-release powders based on polymer particles for pulmonary delivery of beclomethasone dipropionate in the treatment of lung inflamation” deals with the formulation of the mentioned corticosteroid within polymeric particles to be administered as dry powder by using powder inhaler devices. The aim of the work is very interesting. The manuscript  should improve

The following comments and suggestions might help to improve the paper

-          Introduction

Considering that this is a research work focused on the development of drug loaded polymeric particles based on a particular material, the introduction should include more information about this issue. Data on the polymeric material and previous use in the pharmaceutical field is of interest. Scientific literature documenting this topic should be included and commented.

-          Methods and Results

o   The procedure to prepare the nanoparticles loaded with the drug included into the cyclodextrin is not clear. The autor say that the drug was complexed with the cyclodextrin to obtain the CI. Then CI concentration at 0.4% w/v (line 168) was used for spray drying. In the results section , table 3 shows DL %  and EE % values of 2.1% and 82%, respectively for CI Micro. ¿Is this EE value for the CI or for the drug? Which drug amount was considered as the inital drug amount to estimate EE? This point needs clarification. It would be interesting to know the DL and EE for the drug in the cyclodextrin complex and then the final DL and EE in the nanoparticle powder

o   For the quantification of released BDS in artificial mucus the samples were freeze-dried and treated with the same volumen of methanol before HPLC quantification. Same volumen of methanol ¿compared with?. Why freeze-drying??? Please explain

o    Two paragraphs in Result section (lines 383-394 and lines 419-440) should be removed since these are a replication of what is read in the Methods section

o    A brief description of de product used as reference (Clenil®) should be included in order to understand the differences in BDS release curves. What type of formulation is Clenil?

o   The profiles obtained for the assayed polymeric nanoparticles did not show a more sustained release, compared to Clenil, but a higher fraction released for the same period of time. Since Clenil is a comercial product of proven efficacy BDS levels obtained with this medication must be appropriate. Polymeric nanoparticles produced  BDS released concentrations 3 times higher than Clenil. Would this be safe?? Longer period of drug release was  desirable instead of higher drug amount release

o   The Franz-type difusión cell does not seem to be the most suitable experimental model for  the lung. Justify the choice of this in vitro model

Author Response

Answers to Review 2

-Introduction- Considering that this is a research work focused on the development of drug loaded polymeric particles based on a particular material, the introduction should include more information about this issue. Data on the polymeric material and previous use in the pharmaceutical field is of interest. Scientific literature documenting this topic should be included and commented.

As the reviewer suggests, the authors have included a paragraph concerning the polymeric material used to produce the described particles. For this reason, the authors have added the following sentences at page 3, lines 3-11, of the revised version of the manuscript: “An amphiphilic fluorescent derivative of α,β-Poly(N-2-hydroxyethyl)-D,L-aspartamide (PHEA) was selected as the starting polymeric material to produce the particles. In particular, the structural and functional properties of PHEA have been modulated by functionalization with suitable amounts of polylactide (PLA) and polyethylenglycol (PEG). The potential applications of amphiphilic derivatives based on PHEA/polyesters/PEG in the biomedical field, ranging from the production of biocompatible and biodegradable nano- and microparticles for drug delivery and theranostics, up to the application in tissue engineering have already been widely demonstrated in the last twenty years [18–21].”

- Methods and Results -The procedure to prepare the nanoparticles loaded with the drug included into the cyclodextrin is not clear. The autor say that the drug was complexed with the cyclodextrin to obtain the CI. Then CI concentration at 0.4% w/v (line 168) was used for spray drying. In the results section , table 3 shows DL %  and EE % values of 2.1% and 82%, respectively for CI Micro. ¿Is this EE value for the CI or for the drug? Which drug amount was considered as the inital drug amount to estimate EE? This point needs clarification. It would be interesting to know the DL and EE for the drug in the cyclodextrin complex and then the final DL and EE in the nanoparticle powder

      Regarding the reviewer request to clarify the production of the particles loaded with the inclusion complex between BDP and HP-b-Cyd, the authors have revised the reported % concentrations of the components of the feed fluid in the table 1, as some of them were inserted in the first version of the manuscript as incorrect values. The authors apologize and realize that the reduced clarity for the reviewer was generated precisely by the incorrect data reported. Therefore, they have correct them accordingly.

Moreover, the authors would to better explain the steps to obtain and characterize the CI. in detail, the CI was produced as largely described in literature [Nikouei et al, J Incl Phenom Macrocycl Chem, (2012), 72:383-387]; after production and freeze drying, the amount of BDP in the freeze-dried CI was determined by HPLC, useful to calculate the stoichiometric ratio between the drug and the HP-b-Cyd (equal to 1:1). The DL% and the EE% of the CI Micro, reported in table 3, were also calculated by HPLC and expressed as the weight ratio between the entrapped BDP and the CI-Micro. Therefore, these values refer to the drug loaded into the particles as CI. Considering that the referee request arises from the unclear text, the authors have inserted sentences to make the meaning of each value clearer to the readers: at page 4, lines 27-30: “Moreover, the filtrate was freeze-dried and the DL% (expressed as the weight percent ratio between the loaded BDP and the dried CI) was quantified by dispersing the sample in MeOH. After centrifugation and filtration, the supernatant was analyzed by HPLC.”; at page 10, lines 11-12: “corresponding to a DL% (i.e., the weight percent ratio between the drug amount found and the CI) equal to 25 wt%.”; at page 12, lines 29-31: “To calculate the EE% of the CI-Micro sample, the theoretical drug amount present in the CI (stoichiometric ratio between BDP and HP-b-Cyd equal to 1:1) added to the liquid feed was considered.”

o   For the quantification of released BDS in artificial mucus the samples were freeze-dried and treated with the same volumen of methanol before HPLC quantification. Same volumen of methanol ¿compared with?. Why freeze-drying??? Please explain.

      Regarding the clarification requested by the referee on the artificial mucus release test, the authors want to clarify that the fluid withdrawn from the acceptor compartment is DPBS (and not artificial mucus, which instead is used to disperse the particles and placed in the donor compartment); all aliquots of acceptor compartment withdrawn are lyophilized and reconstituted again in MeOH this being an excellent solvent for the BDP, but not for the other constituents of the acceptor medium, which are separated. Therefore, the methanolic solution was analysed by HPLC to quantify the released drug.

o    Two paragraphs in Result section (lines 383-394 and lines 419-440) should be removed since these are a replication of what is read in the Methods section

      Regarding the referee's request to remove parts of the text in Results and discussion, the authors have modified the text of the first paragraph (lines 389-394 of the old version of the manuscript) and cut the overlapping parts with the methods section. However, the authors do not agree to remove the second paragraph (lines 419-440 of the old version of the manuscript) because it contains additional data and new comments (i.e. concerning the obtained mean geometric diameter values) with respect to the methods section.

o    A brief description of de product used as reference (Clenil®) should be included in order to understand the differences in BDS release curves. What type of formulation is Clenil?

      As the  reviewer requests, the authors have inserted a brief description of Clenil at page 15, lines 16-17, of the revised version of the manuscript: “…a nebulizer suspension of BDP, administered to reduce swelling and irritation in the walls of the airways, thus alleviating breathing problems i.e. in patients with asthma.”.

o   The profiles obtained for the assayed polymeric nanoparticles did not show a more sustained release, compared to Clenil, but a higher fraction released for the same period of time. Since Clenil is a comercial product of proven efficacy BDS levels obtained with this medication must be appropriate. Polymeric nanoparticles produced  BDS released concentrations 3 times higher than Clenil. Would this be safe?? Longer period of drug release was  desirable instead of higher drug amount release

       The graph showing the release of the drug from the various formulations demonstrates that, for the same administered dose, the quantity of drug which passes into the biological fluids could be potentially greater when the drug is incorporated into the polymeric particles (and even more so when it is in the form of an inclusion complex). This is very important if we consider that one of the problems of administering inhaled corticosteroids is the low dissolution of the latter in the pulmonary fluid and therefore the obtained low bioavailability. Furthermore, the formulation that is not absorbed is eliminated and often ingested causing undesirable effects. Moreover, being higher the released dose from the polymeric formulation than from the commercial dosage form, at the same starting dose, it is possible to administer a smaller dose of drug to have the same effect as the commercial formulation, which in the case of the particles described in this work could be achieved by reducing the amount of powder to be inhaled.

o   The Franz-type difusión cell does not seem to be the most suitable experimental model for  the lung. Justify the choice of this in vitro model

      Regarding the referee's consideration on the inappropriate choice of Frantz cells as the method of choice to evaluate the diffusion/dissolution of the drug at the pulmonary level, the authors do not agree as many authors have used this method for the same evaluation [Alp and Aydogan, Lipid-based mucus penetrating nanoparticles and their biophysical interactions with pulmonary mucus layer, eur J Pharm Bioph 149 (2020) 45-57; Eedara et al., Dissolution and absorption of inhaled drug particels in the lungs, Pharmaceutics 14 (2022) 2667; Asmawi et al, Size-controlled preparation of docetaxel- and curcumin-loaded nanoemulsions for potential pulmonary delivery, Pharmeceutics 15 (2023) 652; Oh et al., Preparation of inhalable N-acetylcysteine-loaded magnetite chitosan microparticles for nitrate adsorption in particulate matter, int J Pharm 630 (2023) 122454; Michels et al., Nasal administration of a temozolomide-loaded thermoresponsive nanoemulsion reduces tumor growth in a preclinical glioblastoma model, J Controll Release 355 (2023) 343-357]. For dis reason, they have added a significant bibliographic note in the text [Preparation of inhalable N-acetylcysteine-loaded magnetite chitosan microparticles for nitrate adsorption in particulate matter, int J Pharm 630 (2023) 122454].

Reviewer 3 Report

In this paper, the authors prepared controlled-release beclomethasone particles (alone or with cyclodextrin) by spray drying. The topic is interesting, the methods are clearly described, and the results are promising. Therefore, in my opinion, the paper can be published after some revisions.

There is a typographical mistake in the number of pages (top of the page on the right): it is always inserted 17 of 18.

In the introduction, a broader study of the literature has to be performed.

Please consider literature papers regarding the attainment of beclomethasone particles by spray drying, highlighting the limits of previous publications.

Inclusion complexes have been obtained considering innovative techniques, such as supercritical fluids-based processes (see, for example, doi: 10.1016/j.ejpb.2006.11.013; doi: 10.1016/j.jcou.2020.101397).

Author Response

Answer to Review 3

There is a typographical mistake in the number of pages (top of the page on the right): it is always inserted 17 of 18.

Concerning this typing mistake, the editors must correct it.

In the introduction, a broader study of the literature has to be performed.

As the reviewer suggests, the authors have expanded the introduction taking into account the most significant literature found on the topic, inserting several bibliographic notes and the following sentences at page 2, lines 30-46: “In literature, numerous attempts to improve the bioavailability of beclomethasone dipropionate (BDP) to the lungs have been already reported, i.e. by modifying the chemi-cal-physical properties of the BDP-based powder, such as crystallinity percentage and/or drug dissolution rate [18]. In this context, satisfying results have been obtained by producing BDP nanoparticles via supercritical fluids or nanoprecipitation [19–22]. Moreover, additional advantages, such as controlled release and drug protection, have been obtained by loading the drug into nanostructured polymeric and/or lipid systems [23–27]. More re-cently, numerous BDP-based composites have been made as dry powders to be inhaled through innovative devices, to improve at the same time the efficacy of the drug and the aerosolization properties of the formulations. Therefore, different excipients, such as sugars, polyalcohols, polymers and amino acids, have been used to produce inhalable particles via techniques such as supercritical fluids and spray drying (SD) [18,28–30]. In this context, some authors reported the production of inhalable matrices via supercritical assisted atomization with the aim of increasing the dissolution of BDP in biological fluids, i.e. by using cyclodextrins [31]. The obtained particles seem to be very promising as immediate-release BDP formulations for pulmonary delivery, as it was possible to obtain amorphization of the drug and very rapid dissolution kinetics.”

Please consider literature papers regarding the attainment of beclomethasone particles by spray drying, highlighting the limits of previous publications.

As the reviewer suggests, the authors have expanded the introduction taking into account the previous literature found on drug particles obtained by spray drying, inserting several bibliographic notes and the following sentences at page 2, lines 46-53: “Other authors have also showed that the best inhalable BDP particles in terms of aerosol performance are obtained by SD [22]. Recently, BDP-based powder formulations were prepared by SD using different types of lactose carriers and two different dispersion media, which demonstrate that the physicochemical properties of obtained formulations may be easily varied by altering the dispersion media composition [27]. However, in the cited notes no dissolution/ release ki-netics are reported in comparisons with raw drug and/or with formulations in the market.”

Inclusion complexes have been obtained considering innovative techniques, such as supercritical fluids-based processes (see, for example, doi: 10.1016/j.ejpb.2006.11.013; doi: 10.1016/j.jcou.2020.101397).

Regarding the suggestion given by the referee 3 concerning the innovative techniques for the production of inclusion complexes between BDP and cyclodextrins, the authors will keep it in mind for future works.

Round 2

Reviewer 2 Report

I find the revised version appropriate for publication in present form

Reviewer 3 Report

The paper has been improved